# Differential Chemical Profile of Metabolite Extracts Produced by the *Diaporthe citri* (G-01) Endophyte Mediated by Varying the Fermented Broth pH

**DOI:** 10.3390/metabo12080692

**Published:** 2022-07-26

**Authors:** Julio Cesar Polonio, Marcos Alessandro dos Santos Ribeiro, Cintia Zani Fávaro-Polonio, Eduardo Cesar Meurer, João Lúcio Azevedo, Halison Correia Golias, João Alencar Pamphile

**Affiliations:** 1Laboratory of Microbial Biotechnology LBIOMIC/UEM, Departamento de Biotecnologia Genética e Biologia Celular, Universidade Estadual de Maringá, Maringá 87020-900, Paraná, Brazil; marcosquimicauem@hotmail.com (M.A.d.S.R.); cintiazanifavaropolonio@gmail.com (C.Z.F.-P.); halisongolias@utfpr.edu.br (H.C.G.); 2Laboratory of Mass Spectrometry “LabFenn”, Campus de Jandaia do Sul, Universidade Federal do Paraná, Jandaia do Sul 86900-000, Paraná, Brazil; eduardo.meurer@ufpr.br; 3Departamento de Genética, Escola Superior de Agricultura “Luiz de Queiroz”—USP, Piracicaba 13418-900, São Paulo, Brazil; jlazevedo@usp.br

**Keywords:** principal component analysis, endophytes, DI-ESI-MS, multivariate statistics, secondary metabolites

## Abstract

Endophytic microorganisms show great potential for biotechnological exploitation because they are able to produce a wide range of secondary compounds involved in endophyte–plant adaptation, and their interactions with other living organisms that share the same microhabitat. Techniques used to chemically extract these compounds often neglect the intrinsic chemical characteristics of the molecules involved, such as the ability to form conjugate acids or bases and how they influence the solubilities of these molecules in organic solvents. Therefore, in this study, we aimed to evaluate how the pH of the fermented broth affects the process used to extract the secondary metabolites of the *Diaporthe citri* strain G-01 endophyte with ethyl acetate as the organic solvent. The analyzed samples, conducted by direct-infusion electrospray-ionization mass spectrometry, were grouped according to the pH of the fermented broth (i.e., <7 and ≥7). A more extreme pH (i.e., 2 or 12) was found to affect the chemical profile of the sample. Moreover, statistical analysis enabled us to determine the presence or absence of ions of high importance; for example, ions at 390.7 and 456.5 *m*/*z* were observed mainly at acidic pH, while 226.5, 298.3, and 430.1 *m*/*z* ions were observed at pH ≥ 7. Extraction at a pH between 4 and 9 may be of interest for exploring the differential secondary metabolites produced by endophytes. Furthermore, pH influences the chemical phenotype of the fungal metabolic extract.

## 1. Introduction

Endophytic microorganisms are being increasingly studied; of the more than one million published papers, 40,000 have focused on chemically characterizing compounds produced by these organisms [1]. The ability to establish symbiotic relationships with plant tissue and organs in extremely specific microhabitats is frequently provided as the rationale for exploring this ecological niche [2].

Chemical, physical, and genetic factors determined whether or not an endophyte is established, with the endophyte and host plant required to develop a balanced symbiotic relationship [3]. Endophytes can influence plant metabolism, plants can promote microbial metabolism, and both organisms may be responsible for specific metabolic pathways. Furthermore, plants can metabolize compounds produced by endophytes, and vice versa [4]. This complex network of chemical communication and metabolic activity between organisms produces new natural compounds with antifungal, antibacterial, and/or antitumor activities [5,6,7,8,9,10,11,12]. Between 2012 and 2014, Chagas et al. [7] reported more than 300 new metabolic compounds produced by endophytes.

Compounds produced by endophytes include a broad range of structural groups, including benzopyranones, cytochalasins, quinones, steroids, isocoumarins, terpenoids, and others [5,6,13,14,15,16,17,18,19]. Moreover, these microorganisms can produce the same or mimetic compounds derived from the host plant, thereby potentially facilitating the production of metabolic compounds in industrial bioreactors. Indeed, this method has been used to prepare paclitaxel, an antitumor compound produced by *Taxomyces andreanae* isolated from *Taxus brevifolia* [5]. Although properly domesticating endophytes for industrial production remains difficult [20], the discovery of this compound is one of the main reasons for the accelerated exploration of metabolites produced by endophytes.

Endophyte metabolites are routinely studied using axenic endophytic cultures followed by growth in artificial broths and secondary-metabolite extraction by partitioning with organic solvents [1,21,22,23,24,25,26]. The crude extracts are then subjected to analysis and chromatographic separation to produce fractions with higher degrees of purification prior to being chemically analyzed using techniques such as nuclear magnetic resonance (NMR) spectroscopy and/or mass spectrometry (MS) for chemical annotation [23,27,28,29].

Frequent re-isolation and re-characterization of known compounds is a current problem confronted by natural-compound studies. Therefore, new approaches that more easily discriminate between known molecules and new compounds are needed [29]. In addition to chemical-identification techniques, inducing differential secondary metabolic pathways and extraction methods is an approach that can be better exploited [1,4,30,31]. Chemical-composition data can be applied to fungal chemotaxonomic approaches, as reported by Mácia-Vicente et al. [29], who studied the chemical profiles of more than 800 strains, annotated more than 1000 compounds, and identified compounds exclusively found in specific orders.

The solubility of a substance in a solvent is a chemical characteristic that is often overlooked; this qualitative attribute is directly related to molecular structure, especially the polarities of chemical bonds and the chemical species as a whole (dipole moment). Organic compounds may be present as apolar species that are insoluble in water, or they may display alternative polarities and solubilities in various solvents, and at different temperatures and pH values. In addition, functional groups may favor dipole–dipole interactions [32]. Moreover, the molecule may exist as its conjugate acid or base in solution, which influences its solubilities [33]. Consequently, the constitution of a metabolic extract is influenced by pH; consequently, pH is expected to affect the exploration of natural-compound and chemotaxonomic studies.

In this study, we used direct-infusion electrospray-ionization mass spectrometry (DI-ESI/MS) to evaluate how varying the pH following incubation (but before the organic solvent partition step) affects the chemical profiles of endophyte extracts [34], with the data subsequently analyzed using multivariate statistical approaches. This study aimed to demonstrate that the chemical-profile data obtained by DI-ESI/MS show patterns that depend on pH when metabolites are extracted from the same fermented broth using the same microbial strain. This approach may result in liquid–liquid extraction protocols exhibiting higher molecular richness, and provide other poorly soluble molecules using specific solvents and conditions, such as the pH at the end of the fermentation process.

## 2. Results and Discussion

The WV extracts (see Material and Methods section for descriptions of the “WV”, “PV”, and “MV” extract terminology) provided higher yields than the other solvent protocols used in this study (Figure 1), with the highest yields obtained at pH < 7. All treatment protocols provided higher yields than that containing only broth culture medium (blank, B). The mean yield obtained at pH 4 by WV treatment is equivalent to that previously reported [34], demonstrating that the extraction method is experimentally producible.

Metabolomics studies that use mass spectrometry generate immense amounts of raw data, and a substantial portion must be initially filtered to obtain concise data prior to any detailed analyses [35,36]. Therefore, we adopted threshold and exclusion peak lists to filter data obtained by analyzing extracts, which helped to exclude background noise signals and mobile phase peaks. The raw data were processed using the MarkerLynx XS software, which led to 86 ions observed in negative mode and 166 in positive mode.

After normalization using the Pareto scale, the data were subjected to ANOVA followed by Tukey’s honestly significant difference post hoc test using the MetaboAnalyst platform, which led to 140 ions in positive mode and 77 ions in negative mode, with significant differences observed among samples. Three groups were identified based on the sample distribution provided by principal components analysis (PCA) (Figure 2 and Figure 3), namely, B, and samples with pH < 7, and ≥7. Consequently, we confirmed and defined the three groups/clusters using k-means, with samples compared and grouped into these three clusters.

In positive mode (Figure 2), PCA revealed that for principal components PC1—three represent 31.9%, 11.7%, and 10.8% of the variance, respectively. The bi-plot graph shows that some ions, such as those at 257.8 and 358.0 *m*/*z*, are related to PC1 and 2; in contrast, the 226.5 *m*/*z* ion is more representative of PC1. K-means clustering revealed an intraspecific predominance among samples with pH < 7 (Cluster 2) and samples with pH ≥ 7 (Cluster 1), with all B samples well grouped (Cluster 3). PC1 and PC2 revealed that Clusters 2 (mainly pH ≥ 7 samples) and 3 (B) overlap; however, this was overcome by analyzing PC3 (see Appendix A). Nevertheless, two blanks with pH 2 and the PV12 sample remained clustered together with the pH < 7 samples in Cluster 2, and therefore appeared as outliers.

Groups were more distinct in negative mode (Figure 3) than in positive mode. The separation provided by PC1 (with 32.9% of the explained variance) and PC2 (14.2%) enabled k-means to define the clusters well. As demonstrated by loadings analysis, some ions were better a representation for the PCs, namely, those at 203.9 and 240.8 *m*/*z* for PC1, and at 549.2 *m*/*z* for PC2.

Similar results were observed using the k-means approach during hierarchical cluster analysis (HCA) based on the Euclidian distance method and the Ward grouping algorithm (Figure 4). The B2 and PV12 samples were again grouped at pH < 7 in positive mode, with other groups well-resolved.

Minimal differences were observed between WV4, PV2, WV2, and MV4 in negative acquisition mode; however, as was previously observed, three clusters (B, pH < 7, and ≥ 7) appeared in well-defined clades. Overall, when compared to the PCA results, those obtained by HCA show global differences among the three clusters in both positive and negative acquisition modes.

Partial least-squares-discriminant analysis (PLS-DA) provided evidence of possible similarities or specific differences through the preferential organization of PCs that are correlated using classificatory variables of interest (i.e., B, pH < 7, and pH ≥ 7). While this method separated the samples into the above groups in negative mode, HCA revealed a group formed by PV4 and MV2 that is slightly different to the other pH < 7 samples. PLS-DA loading analysis showed that two ions are more differentiated for these samples, namely, those at 150.7 and 240.8 *m*/*z*. This method generated eight components, where the first three had R² and Q² values of 0.76 and 0.61, respectively, using the leave-one-out cross-validation (LOOCV) method. These R² and Q² values indicate that the model exhibits promising data-adjustment and prediction quality [37]. These results were similar to those obtained in positive mode. As was observed using HCA, B2 and PV12 displayed unexpected behavior, with the first three components exhibiting R² and Q² values of 0.81 and 0.62, respectively (see Appendix A).

Appendix A present the top 25 variable influences on projection (VIPs) ions defined by PLS-DA. The chemical profiles of the treated samples acquired in positive acquisition mode are graphically summarized. B2 and PV12 display patterns that are incompatible with the other sample groups, from which we infer that extreme pH variations (i.e., 2 or 12) favor unpredictable chemical reactions; although, we do not have prior knowledge of the exact chemical constitution of the extract [38]. Similar observations are made for other samples at pH 2 and 12, but they are less notable than those made for B2 and PV12. Based on the data acquired using the negative method, we conclude that principal ions are well defined in the pH < 7 samples, but a few discrepancies are observed for the B and pH ≥ 7 samples.

The orthogonal partial-least-squares discriminant analysis (OPLS-DA) approach was applied to reveal the main differences between sample groups generated by k-means (specifically, comparison between Clusters 1 and 2 using both acquisition methods; see Appendix A). The B clusters were not analyzed because we aimed only to identify differences caused by varying the pH. Figure 5A shows the S-plot ion distributions for the Cluster 1 and 2 samples in positive mode, as generated by MarkerLynx XS. The 390.7, 456.5, and 546.5 *m*/*z* ions stand out in the case of Cluster 1, while the others are more notable for Cluster 2. In negative mode (Figure 5B), samples with pH < 7 (Cluster 2) exhibited fewer ions than those in Cluster 1. These findings suggest that the signs of the ions may have intensified and, consequently, they were highlighted as being more statistically important for group separation.

Nielsen and Larsen [33] showed that pH is an important organic-extraction parameter because more ionizing molecules are extracted into the organic phase at neutral pH than charged ones. Furthermore, most fungal metabolites are acidic; hence, a low pH is sometimes required for extraction [39].

Possible favorable chemical reactions are a major problem associated with the addition of acid or base. For example, alcohols can form esters or lactones with carboxylic acids under acidic conditions [38]. For this reason, extracting natural compounds under extreme pH conditions is of little interest because they are unpredictable (as observed for samples extracted at pH 2 and 12).

Figure 5 shows that experimental design is important when acidifying and/or basifying fermented broths prior to organic solvent partitioning. Ions at 390.7 and 456.5 *m*/*z* that appeared mainly under acidic conditions, and ions at 226.5, 298.3, and 430.1 *m*/*z* observed at pH ≥ 7 are the most important differences observed between extracts.

The literature appears to lack studies that consider the pH of the fermented broth when characterizing metabolomic profiles, with process optimization that enables specific compounds to be obtained and purified following fermentation, which is frequently the focus of these studies. For example, Li et al. [40] reported a one-step recovery system for succinic acid from the fermented broth of *Actinobacillus succinogenes* in which adjusting the conditions to acidic pH is a crucial factor in the process because it reduces the solubility of the compound in solution, thereby improving the separation and purification of the product. These achievements rely on the physicochemical characteristics of the molecule of interest, which is present in its acidic form or succinate salt in the fermented broth; the molecule is in its acidic form at pH values below 2.0 and is poorly soluble.

In contrast, studies that evaluate the metabolic profile of a sample using different solvents or a combination of these during extraction are common [41,42,43], either for exploring compounds or even chemotaxonomy purposes [41,42,44,45]. In the latter, the approach adopted in this study can be used to determine the robustness of the chemotaxonomic analysis, considering that different strains of the same species can produce different amounts of various compounds that can alter the physicochemical characteristics of the fermented broth, i.e., pH. Consequently, the genotype of the strain can influence the chemical profile of the extraction product [44].

From the point of view of innovation, new bioactive molecules for medical applications need to be increasingly discovered considering emerging diseases and multidrug-resistant microbial strains [46,47,48]. Therefore, microbial-biomolecule prospecting remains an important niche [47,48]. To this end, non-replicating approaches are increasingly necessary since the quantity and speed at which discoveries are made have both increased due to new methodologies, such as mass spectrometry and its respective databases, which substantially accelerate the ability to chemically characterize compounds [49].

The two metabolic profiles obtained using acidic and basic extraction protocols in this study provide an innovative concept. The data show significant differences that highlight the importance of pH when exploring new biomolecules and facilitate isolation, purification, production, and downstream processes. For example, the ion observed at 430.1 *m*/*z* in positive mode (Figure 5A) was only detected in samples from Cluster 2. Therefore, this compound cannot be extracted under conventional extraction conditions that do not adjust the pH to ≥7.

Other approaches that rely on pH can also be adopted to select target molecules. For example, less molecular diversity is obtained at pH ≥ 7, which is helpful when purifying target molecules. However, yields are also lower, most likely because most metabolites produced by fungi are acidic in nature [33,39].

## 3. Materials and Methods

### 3.1. Endophytic Fungus

The *Diaporthe citri* strain G-01 endophytic fungus was isolated from the leaves of the medicinal *Mikania glomerata* (Spreng.) plant. This microorganism is part of the Collection of Environmental and Endophytic Microorganisms of the Microbial Biotechnology Laboratory at the State University of Maringá (CMEA-LBIOMIC/UEM), Maringá, Paraná, Brazil, and was permanently conserved using the Castellani method.

### 3.2. Fermentation

Fermentation was performed as described in a previous report that determined the optimal conditions for producing 3-nitropropionic acid (3-NPA) in this strain [34]. The fungus was grown on potato dextrose agar (PDA; Acumedia, Michigan, USA) at pH 6.6, for 7 d at 28 °C. Subsequently, 3–4 plugs (6 mm diameter) were inoculated in 500-mL Erlenmeyer flasks with 250 mL of potato dextrose broth (PD; Acumedia, MI, USA) at pH 7.0 and 28 °C for 22 d.

### 3.3. Isolating the Crude Extract after Varying the pH 

After incubation, the mycelia were removed by filtration through a membrane filter using a Büchner funnel and Kitassato apparatus. The pH of the fermented broth was measured after 22 d of fermentation, which resulted in a pH of approximately 4.0. The pH was adjusted with NaOH or HCl solutions according to Figure 6, resulting in 15 treatments referred to as “PV”, “MV”, and “WV” (in addition to blanks [B]). The following experimental design was used: Five samples were subjected to “plus pH variations” (PVs), in which the pH was adjusted to 2.0 followed by liquid–liquid partitioning, which led to the PV2 samples. The pH of the aqueous phase from the same Erlenmeyer flask was adjusted to 4.0, and extracted to give the PV4 extract. This process was repeated until the pH reached 12.0, which resulted in five extracts per Erlenmeyer flask (PV2, PV4, PV7, PV9, and PV12). Five samples were also subjected to “minus pH variations” (MVs) in a similar manner to that described for the PVs, with pH adjusted from 12.0 to 2.0. Lastly, another five samples were not pH adjusted (“without pH variation”, WV) where the pH of each Erlenmeyer flask was individually adjusted to 2.0, 4.0, 7.0, 9.0, and 12.0 and then partitioned, resulting in one extract per Erlenmeyer flask. Therefore, a single flask was sufficient to extract each of the PV and MV samples, while the WV samples required one flask per extraction. “Blank samples” (B) were prepared using PD broth devoid of inoculum and maintained under the same incubation conditions as the fermented broths. B extraction was performed under the same conditions as those used for the WV samples. All extracts were obtained in duplicate.

Liquid–liquid extractions were performed using ethyl acetate (1:5 ethyl acetate/broth ratio) in a separating funnel with 250 mL of pH-adjusted fermented broth. This step was repeated four times. The extracts were combined and the solvent was evaporated using a rotary evaporator (Tecnal TE-210) at 40 °C. Yields were variance-analyzed and Skott–Knott tested (*p* < 0.05 was considered to be significant) to determine the best extraction conditions.

### 3.4. Sample Preparation

The concentration of each extract was adjusted to 1 mg/mL with HPLC-grade methanol. For electrospray ionization (ESI), each sample was diluted using 10 vol% methanol with either 1% ammonium hydroxide (negative mode) or 1% formic acid (positive mode). Each sample was filtered through a 0.45 µm filter and then directly injected (5 µL) into the mass spectrometer. Each sample was analyzed in triplicate.

### 3.5. Instrumentation

Direct infusion electrospray mass spectrometry (DI-ESI/MS) was performed using a PREMIER/XE^®^ Quattro MicroTM API. Data acquisition was controlled using MassLynx version 4.1 software. Prepared samples were injected directly at room temperature into a Rheodyne mass spectrometer. The total run time was 2 min. The desolvation gas-flow rate and source block temperatures were set to 350 and 110 °C, respectively. The desolvation gas-flow and cone gas-flow rates were set to 500 and 0.0 L h^−1^, respectively. Nitrogen gas was used as the drying gas and for misting.

### 3.6. Data Processing and Multivariate Statistical Analysis 

The raw data were first treated using MarkerLynx version 4.1, where filters were used to remove ions from the mobile phase as well as background noise. Specifically, we used an exclusion list obtained from the mobile phase and a threshold of approximately 10^4^ times less than the base peak. The filtered ions were normalized using a Pareto scale. The data were evaluated by analysis of variance (ANOVA; *p* < 0.01 was considered to be significant) and multivariate statistics with unsupervised and supervised approaches: principal component analysis (PCA), hierarchical and non-hierarchical cluster analysis (HCA and k-means, respectively), partial-least-squares discriminant analysis (PLS-DA), orthogonal partial-least-squares discriminant analysis (OPLS-DA), and heatmaps, using the MetaboAnalyst version 4.0 online statistical platform [35] and MarkerLynx XS version 3.0.1.0.

## 4. Conclusions

We demonstrated that the pH of the fermented broth affects the chemical profile of the metabolic extract. In addition, an extreme pH (i.e., 2 and 12) can highly expressively modify the chemical constituents of a sample in an experimentally inconsistent manner that generates outliers. This study revealed the potential of using a pH gradient (between 4 and 9) to explore secondary fungal metabolites once the analyzed samples had formed clusters based on the pH of the fermented broth (pH < 7 and ≥7). Therefore, we propose that extractions at various pH values can provide greater molecular richness in liquid–liquid extraction protocols, affording other molecules that are poorly soluble in a specific solvent under determined conditions, such as the final fermentation process, which may lead to new approaches for exploring fungal metabolites that use this extraction technique.

## Figures and Tables

**Figure 1 metabolites-12-00692-f001:**
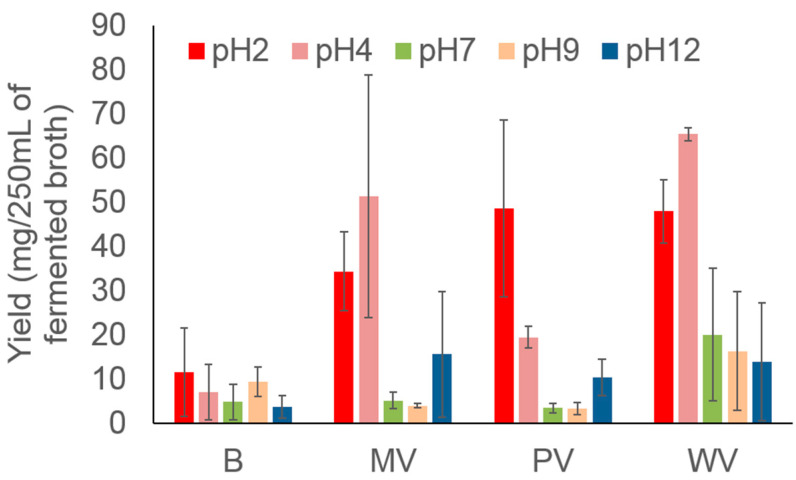
Yields of secondary metabolites extracted using the pH-variation method.

**Figure 2 metabolites-12-00692-f002:**
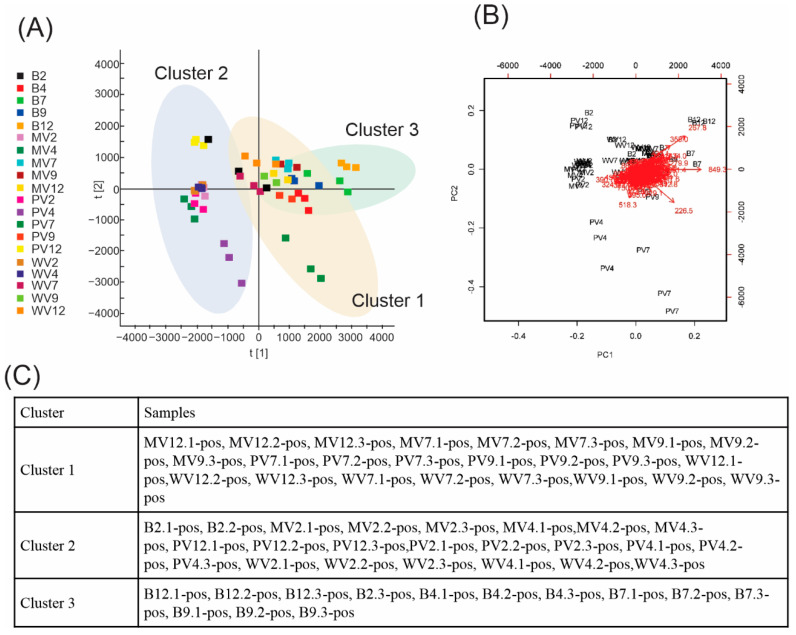
Unsupervised analyses of data acquired in positive mode: (**A**) Principal components analyses (PCAs)—symbols within the delimited transparent areas are those that form clusters defined by k-means. (**B**) Bi-plot analysis (scores × loadings). (**C**) K-means clusters.

**Figure 3 metabolites-12-00692-f003:**
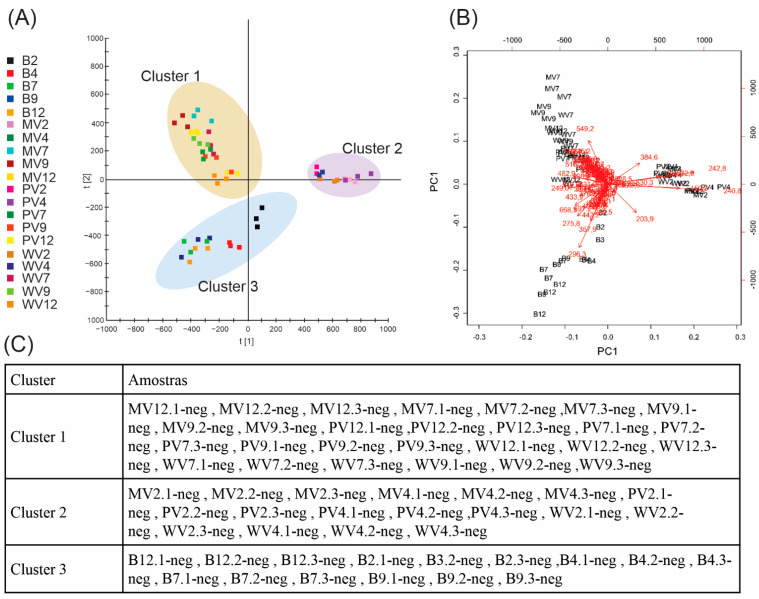
Unsupervised analyses of data acquired in negative mode: (**A**) Principal components analyses (PCAs)—symbols within the delimited transparent areas are those that form clusters defined by k-means. (**B**) Bi-plot analysis (scores × loadings). (**C**) K-means clusters.

**Figure 4 metabolites-12-00692-f004:**
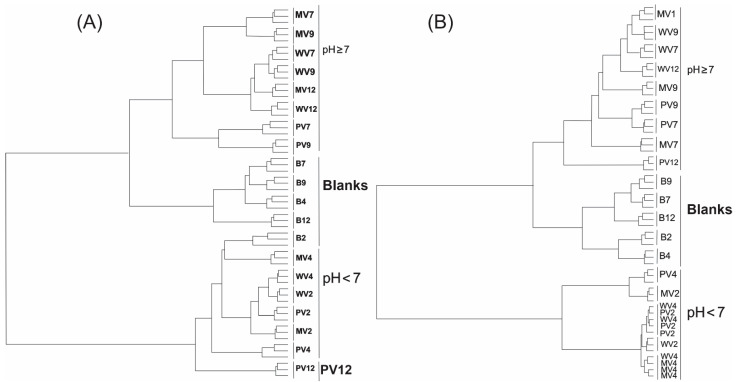
Hierarchical clustering analysis (HCA). Dendrograms were generated by Euclidian distancing and the Ward grouping algorithm for (**A**) samples obtained in positive acquisition mode and (**B**) samples obtained in negative acquisition mode.

**Figure 5 metabolites-12-00692-f005:**
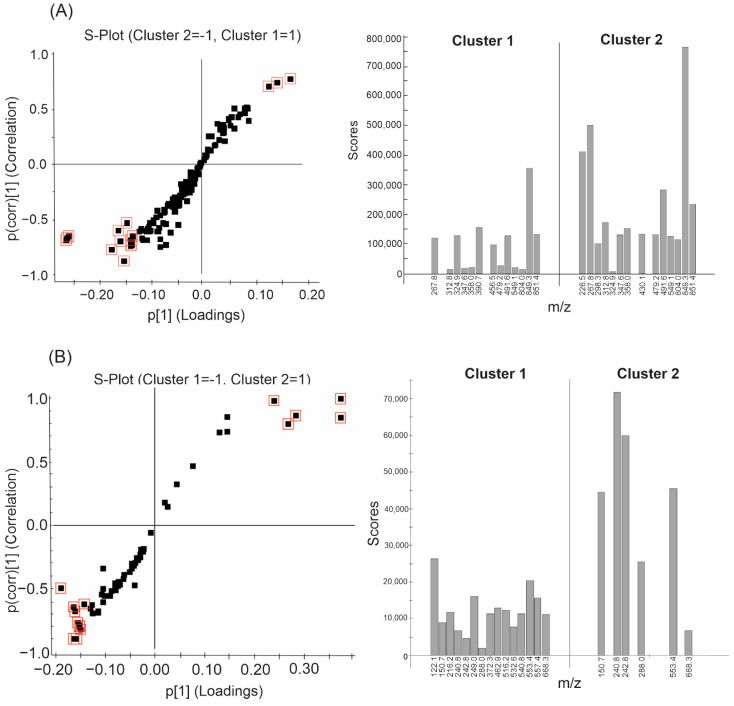
S-plots (left) and normalized mean ion intensities (right) for ions selected in (**A**) positive mode and (**B**) negative mode. The highlighted squares correspond to ions with the highest correlation values (VIPs) in their respective clusters; they are highlighted in the histograms to the right of each S-plot. Ions that are repeated in both samples but stand out in relation to their observed intensities are observed in both clusters. Cluster 1: pH < 7; Cluster 2: pH ≥ 7.

**Figure 6 metabolites-12-00692-f006:**
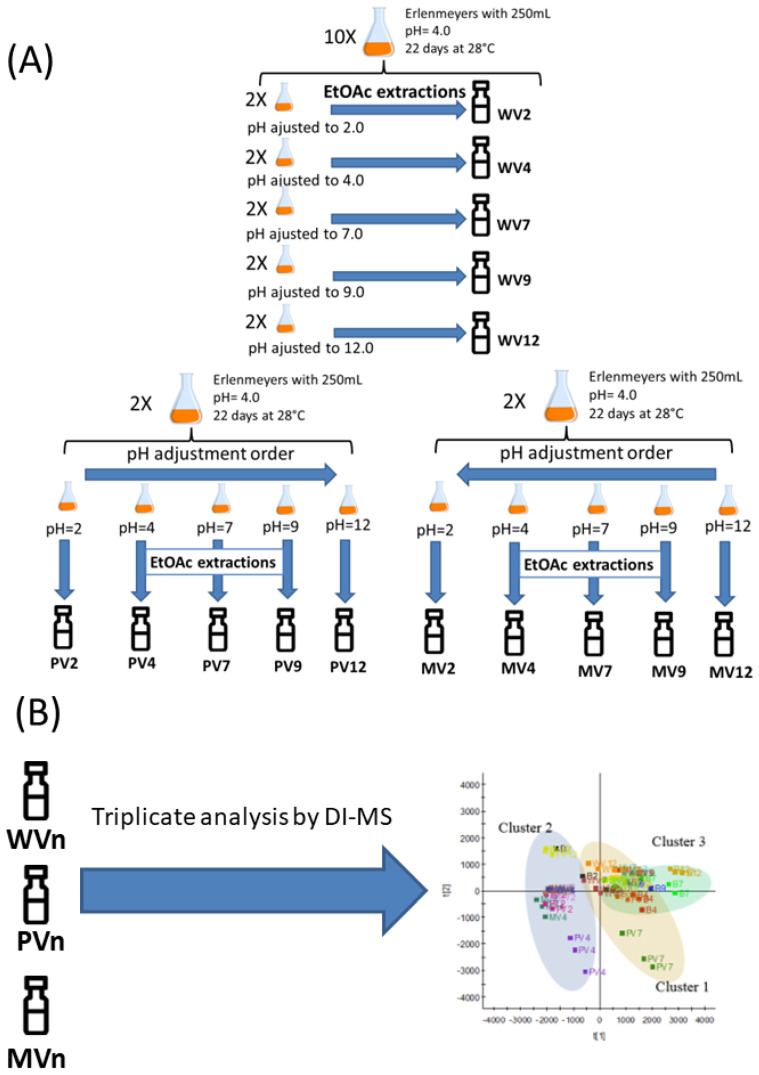
Experimental design: (**A**) Varying the pH of fermented broth. WV samples: the pH of the fermented broth was adjusted, after which the adjusted broth was extracted with ethyl acetate extraction; the broth was discarded after partitioning. PV and MV fermented broths: pH was adjusted to 2 and 12, respectively, after which they were extracted with ethyl acetate after adjusting the pH to 4 or 9, respectively, without discarding the broth. (**B**) The obtained vials (two per sample) were analyzed by DI-MS in triplicate.

## Data Availability

The data presented in this study are available in Appendix A at [doi:10.5281/zenodo.6613511].

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
