# Peer review of "Differential Chemical Profile of Metabolite Extracts Produced by the Diaporthe citri (G-01) Endophyte Mediated by Varying the Fermented Broth pH"

_metabolites, 2022, doi:10.3390/metabo12080692_

Round 1

Reviewer 1 Report

Comments on the ms entitled “Differential chemical phenotype of metabolic extracts pro-2 duced by the endophyte Diaporthe citri (G-01) mediated by var-3 iation in fermented broth pH”

This paper deals with an interesting aspect of research on endophytes and their role in fungus-plant interactions that can stimulate the synthesis of biologically active secondary metabolites. The extraction conditions of these compounds are extremely important to preserve their properties. Despite the novel and interesting research, the presentation of results and discussion needs additions and improvements. I have included comments and concerns in part in the text of the article and the review text.

Here are my comments:

1.       Lines 93-119. My impression is that the Authors of the paper explain and justify the use of multivariate methods. This is evidenced by the works cited. I am well aware of these methods and appreciate their usefulness in the research conducted. In my opinion, the focus should be on the discussion of the results obtained and their discussion.

2.       The authors in my opinion have concerns about the static methods used. They discuss these during the presentation of the results.

3.       My first question about the presentation of the test results in Figures 1-4. Does color matter? Authors use color highlighting of individual samples e.g. B2 black square and B4 red in Fig. 1 A (side of figure). What was the key to color selection and does it matter? No description below the figure. Then there are the colored transparent markings of the different clusters, do these correspond to the different pH values?

4.       Markings on figures illegible. They are difficult to analyze.

5.       The same problem is with the other figures. There are no detailed descriptions of what can be seen on them and how the analyzed results are labeled. In my opinion this should be standardized and explained.

6.       Besides, I miss the comparison of the obtained results with other works on this topic.

Reviewer 2 Report

Sorry to say I reject this manuscript for the following reasons:

1.       There are a lot of wrong sentences that makes following the manuscript hard e.g., lines 87.

2.       Abstract does not contain any important data and I have no idea which pH is good to extract and what compounds affected.

3.       There are no names of compounds that the fungus produces and that they found to be affected by pH.

4.       The manuscript is full of abbreviations that make it confusing for readers to understand e.g., line 86 (abbrev. WV), Line 106 (abbrev. PCA, HCAPLS-DA. The same can be said about the Figures.

5.       The Materials and methods section is hard to understand exactly what was done and how they did it.

6.       Conclusion has nothing to lead you to main results and significant of them.

Based on all above, in my opinion this manuscript, has no scientific value and I reject it for publication.

Round 2

Reviewer 1 Report

I thank the Authors of the paper for taking into account my comments and making corrections to the work. 

Reviewer 2 Report

I reviewed the manuscript again and put down some notes below:

Line 23- what does he word chemical phenotype mean here? Do they mean profile?

Line 21 formed clusters- please explain what clusters are?

Line 23-24: statistical analyses allowed us to determine the presence or absence of high importance values? Can you give an example?  just write after the sentence e.g------- and give your important results.

Line 63 sentence is not correct where starts with Thus,.

Line 92. I think it is bad idea to start the results with taking about a supplementary figure. I suggest Figure S1 become regular Figure that I can see.I did not see Fig. 1 and I don't know how big it is.

Line 100 New sentence starting with Thus, is a wrong sentence that needs correction.

Line 141. What does it mean unsupervised analysis?

Line 296 submitted for pH evaluation. Does this mean they measured pH and was 4.0 and why they need reference? please make it clear.

Line 298 sentence is wrong.

Line 299 they call some samples without pH variation (WV) yet in the next line they say pH was adjusted as in Fig. 5? And the Figure shows they changed the pH to 2,4,7,9 and 12. I don’t understand why they say without variation (WV).  Really this needs good rewording and make it clear to reader.

In Fig 3. They also talk about blank samples but in the materials and methods they never say anything about which samples are Blank?

They talk about pH adjustment but in materials and methods they never say how they did that. What solutions did they use and what pH meter did they use???

Line 354 Were-----? Incorrect sentence.

Line 354 again I don’t understand the word chemical phenotype here. Do they mean chemical profile?
